

# Self-efficacy, relationship satisfaction, and social support: the quality of life of maternal caregivers of children with type 1 diabetes

Einar B. Thorsteinsson, Natasha M. Loi and Kylie Rayner

Department of Psychology, University of New England, Armidale, Australia

## ABSTRACT

**Objectives**. To examine maternal functioning and wellbeing as important aspects of a family's adaptation to chronic paediatric conditions, in particular, children with diabetes.

**Method**. This cross-sectional study investigated the difference between the perceived quality of life of mothers of children with diabetes ($n = 63$) and mothers of children without diabetes ($n = 114$). The study also examined the role of self-efficacy, relationship satisfaction, number of social support providers, and satisfaction with social support in predicting quality of life.

**Results**. Mothers who had a child with diabetes had lower quality of life measured by general health, vitality, social functioning, role-emotional, and mental health than mothers that did not have a child with diabetes. Self-efficacy, relationship satisfaction, and social support were significant predictors of quality of life (mental health domain).

**Conclusion**. In order to enhance their psychological wellbeing, mothers of children with diabetes require adequate psychosocial support. Other implications for research and potential interventions are discussed.

## INTRODUCTION

Diabetes is a serious and rapidly escalating global health problem. In Australia, over 6,000 children under 15 years of age had type 1 diabetes in 2013, representing 139 cases per 100,000 people (*Australian Institute of Health and Welfare [AIHW], 2015*). Type 1 diabetes is a disorder that mainly affects children and adolescents and is directly caused by immune system associated destruction of cells in the pancreas (*Atkinson, Eisenbarth & Michels, 2014*). Diagnosis is based on blood glucose levels or an oral glucose-tolerance test (*Atkinson, Eisenbarth & Michels, 2014*). A diagnosis of type 1 diabetes has life-changing effects on families where a family member, most often the mother (e.g., *Jaser et al., 2009*; *Kovacs et al., 1990*), will find herself in the role of caregiver. The caregiver role will increase daily mental pressures on the caregiver as they have to manage their child's behaviour and health-related factors to a larger extent than before the diagnosis. This can include

Corresponding author
Einar B. Thorsteinsson,
ethorste@une.edu.au

overseeing the child's activity levels (e.g., increase exercise levels and reduce sedentary behaviour), imposing dietary restrictions (e.g., carbohydrate levels), ensuring glycaemic control, and monitoring for signs of hypoglycaemia and hyperglycaemia (e.g., *Smart, Aslander-van Vliet & Waldron, 2009*). Furthermore, the caregiver role varies depending on the age of the child and age of diagnosis varying the demands on the caregiver (*Smaldone & Ritholz, 2011*; *Whittemore et al., 2012*). With the diagnosis, and these new roles for the mother, comes worries (e.g., "will I manage to keep my child safe", "will the disease cause irreparable damage") that may affect mental and physical health (*Lindström et al., 2017*). Therefore, it is imperative that the caregiver has access to and receives satisfactory levels of social support (e.g., emotional, financial, belonging, and informational) from a strong social support network in order to attenuate the effects of caregiver stress on the caregiver's mental health. The present study concentrates on type 1 diabetes and considers the quality of life of maternal caregivers and the psychosocial resources that may play a part in determining their perceived quality of life.

The difficulties in balancing the child's psychological needs in conjunction with the restrictions and treatment requirements imposed by diabetes can lead to frustration in the caregiver (e.g., *Cunningham et al., 2011*; *Monaghan et al., 2009*). Diabetes can become burdensome, place a strain on financial resources, reduce the enjoyment of the parental role, and possibly impact upon both the physical and mental health of the caregiver (*Helgeson et al., 2012*; *Mellin, Neumark-Sztainer & Patterson, 2004*).

Measuring the mothers' quality of life captures the broad impact of the child's illness on the mother's physical, psychological, and social wellbeing and functioning (*Quittner et al., 1998*). A poorer quality of life among caregivers has been reported among (a) parents of children with Down's syndrome (*Hedov, Anneren & Wikblad, 2000*), (b) families of children with autism spectrum disorder (*Gardiner & Iarocci, 2015*), and (c) mothers of children with (i) leukaemia (*Yamazaki et al., 2005*), (ii) asthma (*Cerdan et al., 2012*), and (iii) cerebral palsy (*Eker & Tuzun, 2004*).

Research has indicated that parental psychological adjustment influences the adjustment of the chronically ill child (*Muscara et al., 2015*; *Wagner et al., 2003*). Increased distress among mothers of children with type 1 diabetes has been shown to predict concurrent child psychological distress (*Lopez et al., 2008*). Increased depressive symptoms in children (aged 10–18) with diabetes have been associated with diabetes-related caregiver burden and diabetes-related family conflict (*Hood et al., 2006*).

In view of the differences in individuals' perceived quality of life, it is appropriate to consider maternal resources that could be utilised to improve physical and mental health. Elements of the transactional stress and coping (TSC) model of adjustment to chronic illness (*Hocking & Lochman, 2005*) provide a theoretical basis for this investigation. In the context of the present study, the TSC model suggests that type 1 diabetes is a stressor the caregiver has to adapt to (i.e., learn to cope and live with). The factors that affect the caregiver's ability to adapt to their role are varied and can include the caregiver's social support levels, relationship with their partner, general self-efficacy, coping strategies, stress appraisal, and socioeconomic status (*Azar & Solomon, 2001*). The present study focuses on aspects of the TSC model as supported by the literature reviewed. Studies from parental

wellbeing literature identify self-efficacy as one of the personal resources associated with improved quality of life. General self-efficacy refers to confidence in the ability to perform the necessary behaviours to influence or control the forces affecting one's life in order to attain a certain outcome. Greater psychological wellbeing is associated with higher levels of self-efficacy (*Eiser et al., 2001*). Self-efficacy is an important characteristic in alleviating the impact of psychological stress (*Chemers, Hu & Garcia, 2001*) and in general parenting literature, parents' self-efficacy has been reported to moderate the effects of stress on the parents' mental health (*Kwok & Wong, 2000*). In research specific to diabetes, lower self-efficacy in parents was associated with heightened levels of stress and anxiety (*Streisand et al., 2008*). As parents of children with diabetes tend to be actively involved in the process of managing their child's illness, success in controlling the symptoms of the disease can result in heightened self-efficacy which may assist in maintaining the parents' stable and emotional physical state (*Lowes, Lyne & Gregory, 2004*).

A second resource that may affect perceived quality of life is the quality of the parents' marital/de-facto relationship. In a study involving the general population, individuals in discordant relationships reported higher levels of general distress and poorer perceived health (*Whisman & Uebelacker, 2006*). Marital distress can also have a detrimental impact on children, increasing the risk for poorer psychosocial adjustment (e.g., *Repetti, Taylor & Seeman, 2002*). The presence of diabetes in a child has been associated with reduced family functioning (*Crain, Sussman & Weil Jr, 1966*; *Popp et al., 2014*). These findings are compounded as significantly better diabetic control in children has been found when their mothers report marital satisfaction (*Marteau, Bloch & Baum, 1987*).

Another resource that factors prominently in understanding parents' quality of life is the parents' perceived social support levels. Studies indicate that social support has beneficial effects on psychological wellbeing (*Siedlecki et al., 2014*). Larger social support networks and greater satisfaction with social support were significantly associated with better psychological adjustment and reduced distress in mothers of children with chronic physical conditions (*Horton & Wallander, 2001*). Social support from the partners of mothers with adolescents with type 1 diabetes plays an important role in reducing diabetes-related conflict between mother and adolescent, and increases the adolescents' adherence to treatment (*Lewandowski & Drotar, 2006*). These findings have been supported with support for the caregiver being related to better illness management (*Carcone et al., 2011*). Given the positive effects social support seems to have on conflict, illness management, and adherence, increased support may potentially contribute to a reduction in mental health strain experienced by the caregivers.

The purpose of the present study was to examine maternal functioning and wellbeing among mothers of children with diabetes by comparing them to mothers who do not have children with diabetes. Given the significance of diabetes and the intrusiveness of the treatment requirements, it is not surprising that mothers of children with diabetes are considered at risk for psychosocial and family dysfunction as they are required to adapt to physical, emotional, social, and financial challenges. Research suggests that the psychosocial functioning of mothers of children with type 1 diabetes will differ to that of mothers of children without diabetes. As research has suggested that fathers tend to be

less involved in their child's diabetes management (e.g., *Seiffge-Krenke, 2002*), our study focused solely on mothers. The following hypotheses were tested: (a) maternal caregivers of children with diabetes would be more likely to report a perceived poorer quality of life than mothers without a child with diabetes, and (b) lower levels of general self-efficacy, less relationship satisfaction, less satisfaction with social support, and fewer social support providers would be significant predictors of poorer maternal mental health (the quality of life mental health domain) in mothers. The first hypothesis focuses on quality of life while the second hypothesis focuses on the mental health domain of quality of life given the strong focus on mental health in the literature. Increased burden associated with caregiving affects the caregivers' quality of life which in turn affects their ability to continue in their role as a caregiver and their ability to maintain good mental health.

# MATERIALS & METHODS

## Participants

Mothers of a child with type 1 diabetes ("mothers of a child with type 1 diabetes group", $n = 63$) and mothers without a child with type 1 diabetes ("comparison group", $n = 114$) were recruited (*Thorsteinsson, Rayner & Loi, 2016*). The mothers were aged 25 to 52 ($M = 39.33$, $SD = 5.67$). An invitation to participate in the study was provided to members of area chapters of the Juvenile Diabetes Research Foundation throughout Australia, in an article placed in the Diabetes Australia (NSW) quarterly magazine, "Issues", on an Australian online forum for parents of a child or adolescent with type 1 diabetes, "Munted Pancreas", and through the clinics of various physicians in Queensland who specialise in paediatric endocrinology. The comparison group was recruited through school newsletters, parent online forums, and by word-of-mouth.

Mothers were excluded from participating if they were under 18 years of age or if their child had also been diagnosed with another major illness or shown evidence of a developmental disability.

The mean age of the mothers of children with diabetes and the mothers without children with type 1 diabetes were similar (see Table 1). Most identified as Australian (91%), were currently in a relationship (84%), had attained at least an undergraduate education degree (57%), and reported a relatively high income (61% indicated a gross annual household income over $62,000).

The mean age of the children with diabetes was 9.75 years ($SD = 2.98$, ranging from 4 to 15). Time since diagnosis varied: from one to three month ($n = 4$, 6.3%), three and to six months ($n = 2$, 3.2%), six to 12 months ($n = 9$, 14.3%), from 12 months to two years ($n = 7$, 11.1%), and more than two years ($n = 41$, 65.1%). Children's mental and physical health was good ($M = 7.83$, $SD = 1.58$) based on the question "How would you rate mental and physical health of child" 1 (*poor*) to 10 (*excellent*). Only one child had a rating of 3 while the remainder had ratings of 5 and above. When it came to the mothers' biggest concerns, diabetes was highest for 68.3% ($n = 43$), behavioural issues for 1.6% ($n = 1$), social relationships 11.1% ($n = 7$), and other for 19.0% ($n = 12$).

**Table 1  Comparisons of sociodemographic variables for mothers with and without a child with diabetes.**

| Sociodemographic variable | Mothers of child with diabetes ($n = 63$) | Mothers of child without diabetes ($n = 114$) | Comparison |
|---|---|---|---|
| Mother's age in years, $M$ ($SD$) | 39.76 (5.20) | 39.10 (5.92) | $t(175) = -0.75, p = .456$ |
| Number of children, $M$ ($SD$) | 2.49 (0.93) | 2.44 (1.02) | $t(175) = -0.34, p = .731$ |
| Level of education, $M$ ($SD$) | 4.22 (0.98) | 4.73 (1.02) | $t(175) = 3.22, p = .002$ |
| Household weekly income, $M$ ($SD$, $n$) | 3.53 (1.44, 53) | 4.06 (1.19, 106) | $t(157) = 2.47, p = .015$ |
| Ethnicity, $n$ (%) | | | $\chi^2(1) = 3.54, p = .089, \phi = -.14$ |
| - Australian | 61 (96.8) | 101 (88.6) | |
| - European/ Asian/Other | 2 (3.2) | 13 (11.4) | |
| Marital status, $n$ (%) | | | $\chi^2(1) = 1.07, p = .344, \phi = .08$ |
| - In a relationship | 53 (84.1) | 102 (89.5) | |
| - Not in a relation-ship | 10 (15.9) | 12 (10.5) | |
| Employment status, $n$ (%) | | | $\chi^2(4) = 7.36, p = .118$, Cramer's $V = .20$ |
| - Employed part-time | 18 (28.6) | 48 (42.1) | |
| - Student (part or full time) | 4 (6.3) | 8 (7.0) | |
| - Homemaker | 22 (34.9) | 20 (17.5) | |
| - Employed casually | 4 (6.3) | 9 (7.9) | |
| - Employed full-time | 15 (23.8) | 29 (25.4) | |
| Location, $n$ (%) | | | $\chi^2(1) = 4.65, p = .047, \phi = .16$ |
| - Metropolitan area | 41 (65.1) | 91 (79.8) | |
| - Regional area | 22 (34.9) | 23 (20.2) | |
| Family history of diabetes, $n$ (%) | | | $\chi^2(1) = 1.61, p = .261, \phi = .10$ |
| - Yes | 22 (34.9) | 51 (44.7) | |
| - No | 41 (65.1) | 63 (55.3) | |

**Notes.**
Education level: High school, 3; Vocational qualification, 4; Undergraduate, 5; Postgraduate, 6; Household weekly income: up to $400, 1; $401–$800, 2; $801–$1,200, 3; $1,201–$1,600, 4; >$1,600, 5.
Two-tailed $p$ values reported.

## Materials

The demographic questionnaire collected information pertaining to the mother including age, ethnicity, highest level of education achieved, current employment status, marital status, gross household income, residential location, number of children, and any family history of diabetes. Information was also gathered on the demographic characteristics of each of the mother's children including age, sex, whether or not each child had been

diagnosed with diabetes, and if a developmental disability or another chronic illness was present.

The mother's quality of life was measured using the 36-item Short Form Health Survey (SF-36; *Ware et al., 1993*). Thirty-five of the items are aggregated into eight subscale or domain scores: physical functioning, role-physical (i.e., role limitations due to physical health), bodily pain, general health, vitality, social functioning, role-emotional, and mental health. The remaining item was not needed for the present study; it asks respondents the amount of change in their general health over the previous year. There is no total score for the SF-36 and the response format varies greatly. Some items are answered on a 5-point Likert-type scale from *poor* to *excellent*, some items are answered on a 3-point scale from *yes, limited a lot* to *no, not limited at all*, and some on a 5-point scale from *all of the time* to *none of the time*. Each of the domains is attributed a score between 0 and 100, with 100 representing optimal functioning. Internal consistency of each domain ranged from .81 to .88 in the present study.

The General Self-Efficacy Scale (*Schwarzer & Jerusalem, 1995*) measures perceived self-efficacy with the aim of predicting how one copes with daily hassles and adapts to stressful life events. Each of the 10 items refers to successful coping and implies an internal-stable attribution of success (e.g., ''I can usually handle whatever comes my way'') with responses ranging from 1 (*not at all true*) to 4 (*exactly true*). Scores for each of the items are totalled and range from 10 to 40 with a higher score indicating a greater level of general self-efficacy. The present data yielded an alpha of .93.

Relationship satisfaction was measured using the Dyadic Adjustment Scale (DAS; *Spanier, 1976*), a widely utilised and well-validated 32-item measure of marital or partner functioning and satisfaction. The DAS yields a total measure of satisfaction with the relationship as well as four subscale scores assessing dyadic consensus, dyadic satisfaction, dyadic cohesion, and affectional expression. The total DAS score has been suggested as the best measure of dyadic quality (*Sharpley & Cross, 1982*). The response format varies across items. Agreement and disagreement for various items (e.g., friends, religious matters, in-laws) are rated on a 6-point Likert-type scale from *always agree* to *always disagree*. Some items are answered using a *yes-no* or *tick* or *do not tick* dichotomy. Total scores can range from 0 to 140 with higher scores suggesting better relationship satisfaction. The total scale had an internal consistency reliability of .96 in the present study.

The Short Form Social Support Questionnaire (*Sarason et al., 1987*) is a brief measure of social support consisting of items in which a situation is presented and asking for a list of supportive persons (up to 9 per question) and a rating of satisfaction with support ranging from 1 (*very dissatisfied*) to 6 (*very satisfied*). For each individual, the summary measures were the mean number of supporters and mean satisfaction (across all the questions). Higher mean scores in these scales imply a larger support group and greater satisfaction respectively. Cronbach's alpha was .90 for support satisfaction and .94 for support numbers in the present study.

## Procedure

The self-administered questionnaire was available in both pencil-and-paper format and online. Mothers were asked to read the Information Sheet which introduced the study and highlighted the aims. The Information Sheet indicated that by completing and returning the questionnaire, participants were confirming their consent to participate. For those participants who completed the questionnaire online, it was compulsory for each participant to select 'yes' when asked to indicate their consent before they were able to continue on to complete the online questionnaire. A total of 14 questionnaires were returned in the post (nine from the mothers of a child with type 1 diabetes group and five from the comparison group). Participation took approximately 30 min and participants were advised they were able to discontinue at any time without repercussion. Ethics approval was given by the University Of New England Human Research Ethics Committee (HE07/034).

## Statistical analyses

Statistical analyses were completed using SPSS 21. Differences between groups were assessed using: (a) statistical significance employing $t$-tests (*Student, 1908*) a standard test to assess the differences between two independent means and (b) effect sizes employing Hedges' $g$ as it captures effect sizes in standard deviation units (*Borenstein et al., 2009*) and is regularly used when aggregating differences between groups using meta-analysis. Hedges' $g$ also allows for the evaluation of differences between groups through its confidence intervals. Multiple regression analysis (*Tabachnick & Fidell, 2001*) was employed to evaluate different predictors of quality of life. The social support and general self-efficacy measures were not completed by 14 and 17 participants, respectively. The relationship satisfaction questionnaire was not completed by 16 participants who had earlier indicated they were currently in a relationship. As entire measures were incomplete, means were not substituted for missing variables but rather each case with the missing data was excluded from relevant analyses, resulting in different sample sizes for different statistical analyses. A total of 18 participants chose not to disclose household income data.

# RESULTS

## Preliminary analyses

Table 1 shows the differences between mothers of children with diabetes and without diabetes on the demographic variables. Mothers of a child with type 1 diabetes had significantly less formal education than those in the comparison group and they had significantly lower household income. The mothers of children without diabetes tended to reside in metropolitan areas of Australia more than the mothers of children with diabetes.

## Hypothesis 1: perception of quality of life

Mothers of a child with diabetes reported moderately lower mean quality of life scores than mothers without a child with diabetes for general health, vitality, social functioning, role-emotional, and mental health, see Table 2.

**Table 2** Comparison of mothers of children with type 1 diabetes and mothers of children without diabetes.

| Measure | Mothers of child with type 1 diabetes ($n = 63$) | | Mothers of child without diabetes ($n = 114$) | | $t$ (175) | $p$ | Hedges' g |
|---|---|---|---|---|---|---|---|
| | M | SD | M | SD | | | |
| General health | 63.83 | 21.73 | 73.82 | 18.09 | 3.27 | .001 | 0.51 [0.20, 0.82] |
| Vitality | 44.29 | 23.02 | 54.43 | 20.49 | 3.02 | .003 | 0.47 [0.16, 0.78] |
| Social functioning | 70.04 | 25.94 | 80.70 | 21.63 | 2.92 | .004 | 0.46 [0.15, 0.77] |
| Role-emotional | 56.08 | 45.13 | 74.85 | 38.80 | 2.91 | .004 | 0.45 [0.14, 0.77] |
| Mental health | 61.40 | 20.66 | 71.44 | 16.87 | 3.50 | .001 | 0.55 [0.23, 0.86] |
| Physical functioning | 90.08 | 15.41 | 91.05 | 12.61 | 0.45 | .651 | 0.07 [−0.24, 0.38] |
| Role physical | 78.17 | 32.84 | 79.17 | 35.19 | 0.18 | .854 | 0.03 [−0.28, 0.34] |
| Bodily pain | 72.33 | 18.13 | 72.39 | 18.78 | 0.02 | .986 | 0.00 [−0.30, 0.31] |

## Hypothesis 2: predictors of quality of life—mental health

Better mental health was predicted (8% of the variance) by caring for a child without diabetes, living in a metropolitan location, having a partner, higher income, and lower education level, see Model 1 in Table 3 (beta > |0.09|). However, as Table 3 shows (Model 2; beta > |0.09|), the impact of these diminished and reversed (see relationship status) when other measures were added. Model 2 shows that better mental health is predicted by living in a metropolitan location, lower education level, high social support (number and satisfaction), high relationship satisfaction, and high general self-efficacy.

## DISCUSSION

The present findings enhance our understanding of the impact of having a child with diabetes and how maternal quality of life can potentially be improved. The results demonstrate the importance of considering the psychosocial status of the mother when treating a child with diabetes. Effective treatments for children and adolescents with diabetes should include the family, and in particular the mother, as an integral part of the treatment.

Mothers of children with diabetes reported a poorer quality of life than mothers of children without diabetes on five of eight quality of life domains: general health, vitality, social functioning, role-emotional, and mental health. These findings are consistent with findings indicating that parents of chronically ill children, and mothers in particular, are at risk for diminished psychological health (e.g., *Hedov, Anneren & Wikblad, 2000*; *Helgeson et al., 2012*; *Yamazaki et al., 2005*). The current results imply that mothers of children with diabetes are more inclined to evaluate their health as poor and believe it is likely to worsen, more often experience fatigue, tend to have less opportunity for social interactions, have more frequent problems with daily activities as a result of emotional health, and tend to feel more nervous and depressed than mothers without a child with diabetes.

Both groups of mothers had similar scores on the three remaining quality of life domains. These domains have a strong physical component (i.e., physical functioning, role physical

**Table 3  Summary of hierarchical regression analysis of scores on the quality of life mental health domain.**

| Predictor | $B$ | 95% CI for $B$ Lower | Upper | $\beta$ | $r$ | $sr^2$ |
|---|---|---|---|---|---|---|
| Model 1 | | | | | | |
| Mother group | −5.80 | −12.49 | 0.89 | −0.15 | −.18 | .02 |
| Location | −4.38 | −12.25 | 3.49 | −0.10 | −.19 | .01 |
| Relationship status | −20.10 | −44.69 | 4.50 | −0.14 | −.18 | .02 |
| Household income | 2.84 | −0.12 | 5.80 | 0.19 | .23 | .03 |
| Education | −2.06 | −5.24 | 1.12 | −0.12 | −.01 | .01 |
| Model 2 | | | | | | |
| Mother group | 0.81 | −5.13 | 6.75 | 0.02 | −.18 | <.01 |
| Location | −6.21 | −13.08 | 0.66 | −0.15 | −.19 | .03 |
| Relationship status | 8.41 | −14.07 | 30.90 | 0.06 | −.18 | <.01 |
| Household income | 0.91 | −1.63 | 3.45 | 0.06 | .23 | <.01 |
| Education | −2.46 | −5.20 | 0.29 | −0.14 | −.01 | .03 |
| Number of social supports | 1.55* | 0.09 | 3.02 | 0.19 | .41 | .04 |
| Social support satisfaction | 3.61* | −0.30 | 7.53 | 0.18 | .41 | .03 |
| Relationship satisfaction | 0.14 | 0.00 | 0.29 | 0.18 | .36 | .04 |
| General self-efficacy | 1.46* | 0.77 | 2.15 | 0.36 | .45 | .13 |

**Notes.**

$sr^2$, squared semi-partial correlation (squared Part correlation from SPSS); $r$, zero-order correlation.

The quality of life mental health domain is attributed a score between 0 and 100, with 100 representing optimal functioning.

Mother group: 1, mother without a child with type 1 diabetes group; 2, mothers of a child with type 1 diabetes group.

Location: 1, metropolitan; 2, regional.

Relationship status: 1, partner; 2, no partner.

Household income (gross weekly): 1, up to $400; 2, $401 to $800; 3, $801 to $1,200; 4, $1,201 to $1,600; 5, $1,601 or more.

Education: 3, High School (year 10 or year 12); 4, Vocational Training Course/Diploma; 5, Undergraduate; 6, Postgraduate.

Model 1: Adjusted $R^2 = .08$, $F(5, 117) = 3.09$, $p = .012$.

Model 2: Adjusted $R^2 = .36$, $F(9, 113) = 8.53$, $p < .001$.

Change $R^2 = .29$, $F(4, 113) = 13.65$, $p < .001$.

$^*p < .05$.

and bodily pain). As diabetes does not require extraordinary physical exertion on behalf of the caregiver, unlike some other chronic illnesses such as cerebral palsy (*Eker & Tuzun, 2004*) or caring for a child with physical disabilities (*Tong et al., 2002*), it is not surprising that the two groups of mothers had similar scores on these domains.

To understand why some mothers cope better than others, we examined different factors that play a role in the caregiving experience. Examination of the mothers' reports of general self-efficacy, relationship satisfaction, and social support was undertaken to understand further the variables that may be relevant when assessing quality of life among maternal caregivers. As hypothesised, this set of resources accounted for important differences in perceived mental health quality of life. Mothers with a more positive view of their self-efficacy report better functioning corresponding with other findings that have repeatedly shown self-efficacy to be associated with better physical and psychological health (e.g., *Motl et al., 2009*; *Vander Horst et al., 2007*). The findings suggest that mothers who do not feel adequately prepared to handle aspects of their child's diabetes may allow this feeling

of inadequacy to permeate other areas of her life. It is also possible that low self-efficacy decreases the likelihood of the use of appropriate cognitive coping strategies to effectively reduce negative effect. Thus, mothers of a child with diabetes who report poor quality of life could benefit from additional diabetes education and counselling or problem-solving training to boost their confidence and increase self-efficacy. Doing so could help to improve the mother's general wellbeing and ultimately result in better child-health outcomes.

The current findings suggest that having a child who places exceptional demands upon the mother may result in less time and energy for contacts with informal support networks, resulting in a reduction of the number of social support providers and ultimately having a negative impact on maternal quality of life. This suggests that increasing the number of social support providers will likely have a positive impact on quality of life and so increasing the opportunities for expansion of the social support network would be beneficial in improving the mothers' quality of life, particularly in regional areas. Dissatisfaction with the quality of a close dyadic relationship contributed to diminished mental health and wellbeing. Thus it is important that the relationship between parents with children with diabetes be evaluated for poor functioning.

Demographic factors were also found to negatively affect social and emotional functioning, vitality, and mental health (i.e., living in a rural area with a child who has diabetes, not being in a relationship, earning a low income, and having a higher level of education). However, these demographic factors (i.e., location and household income) had smaller effects on quality of life in Model 2. The limited effect for these factor in Model 2 can possibly be attributed to the additional factors in Model 2 such as social support factors, relationship satisfaction, and general self-efficacy. These additional factors seem to supersede the demographics potentially capturing the underlying mechanism responsible for quality of life or capturing an improved assessment of important variables that determine quality of life. Thus, not having a partner negatively affected quality of life in Model 1 while in Model 2 this effect was replaced with relationship satisfaction where the higher the satisfaction the better the quality of life. Location and education did not change much from Model 1 to Model 2 suggesting that a regional location combined with higher levels of education are more detrimental to quality of life than a metropolitan location combined with lower levels of education. This may be caused by the lack of access to services in regional locations putting more burden on caregivers. Furthermore, caregivers with higher levels of education seem to feel the pressures of caregiving more (e.g., *Ory et al., 1999*).

## Limitations and future studies

The data in the present study is cross-sectional thus it cannot demonstrate causal relationships or clarify causal direction. Longitudinal studies that monitor change over time in maternal adjustment with diabetes management, along with repeated monitoring of the child's medical and psychosocial outcomes, could clarify the findings presented here further. Such monitoring might help reveal causal relationships where mothers' status, with or without a children with diabetes, may affect social support, self-efficacy, and relationship satisfaction that, in turn, affects quality of life (see Table 3 Model 2).

Furthermore, additional comparison groups could be included such as mothers of children with asthma or other chronic diseases and questions such as time of diagnosis.

Future research could strengthen the present findings by including medical data pertaining to the child's metabolic control. While it is a limitation of the present study that the severity of the children's diabetes was not assessed, research is inconsistent on the role of disease severity. Some research has shown that illness severity may not play a role in the distress of the caregiver (e.g., *Canning, Harris & Kelleher, 1996*; *Rodrigues & Patterson, 2007*). However, it is possible that illness severity may moderate the pattern of relationships found here, particularly if severity affects the mother's appraisal of the controllability of her child's diabetes, which in turn affects her perception of self-efficacy. Future studies should also consider the effects of sleep disruption in parents of children with type 1 diabetes (*Landau et al., 2014*) given the impact sleep quality can have on health (*Buysse et al., 2010*).

There were significant socioeconomic differences between the two comparison groups (i.e., household weekly income and level of education) which may have resulted in some bias especially in the mental health and social functioning domains of quality of life.

## Conclusion

The present study demonstrates the importance of social support providers, relationship satisfaction, and general self-efficacy to the psychological adjustment of mothers of children with diabetes. Treatment of diabetes in children and adolescents should include close monitoring of the mother's mental health and the provision of appropriate psychosocial support.

### Funding

The authors received no funding for this work.

### Competing Interests

The authors declare there are no competing interests.

### Author Contributions

- Einar B. Thorsteinsson conceived and designed the experiments, performed the experiments, analyzed the data, wrote the paper, prepared figures and/or tables, reviewed drafts of the paper.
- Natasha M. Loi wrote the paper, prepared figures and/or tables, reviewed drafts of the paper.
- Kylie Rayner conceived and designed the experiments, performed the experiments, analyzed the data, wrote the paper, prepared figures and/or tables.

### Human Ethics

The following information was supplied relating to ethical approvals (i.e., approving body and any reference numbers):

The University of New England Human Research Ethics Committee granted approval to carry out the study (HE07/034).

## Data Availability

Thorsteinsson, Einar; Rayner, Kylie; Loi, Natasha M (2016): Children with diabetes: Mothers' quality of life. figshare.

https://doi.org/10.6084/m9.figshare.3208549.v1.

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
