# Peer review of "Self-efficacy, relationship satisfaction, and social support: the quality of life of maternal caregivers of children with type 1 diabetes"

_PeerJ, doi:10.7717/peerj.3961_

## Round 0.1 · original submission · Minor Revisions

All reviewers have made good and thoughtful points regarding the manuscript. which you should attend to. I would particularly draw your attention to the very thorough and detailed comments of reviewer 3.

The impact of the age of the children with diabetes is especially relevant, and should be explicitly discussed in relation to the dynamics of the parental-child relationship. There are also a number of linguistic changes which would make the text more acceptable.

Overall, with careful attention to the reviewers' comments, this paper stands a good chance of publication.

·

Basic reporting

The article is written in English.
The authors have used clear text.

The introduction should include more information, changes in the daily life of caregivers due to the specific characteristics of type 1 diabetes, such as glycemic control, dietary restrictions, physical exercise, and mothers' fear of The disease causes irreparable damage, hypoglycemia, etc.
It should also include a brief description of the disease (type 1 diabetes) and its psychosocial characteristics.

Therefore, the introduction is incomplete

Experimental design

It is not clear in the text the purpose of the study

I consider it important to include in the text the average age of the participants, as well as the standard deviation. Text include age of the child but no of the mothers.

Used instruments are correct, and have high levels of reliability

Methods is described with sufficient information to be repproducible by another investigator.

The time of diagnosis of diabetes in childs has not been taken into account. This variable will affect the results, It would be important to include it in the text.

Validity of the findings

The results are novel and easily replicable by other authors.

Additional comments

The study brings knowledge to science. However, it would be interesting to carry out comparisons between mothers of children with type 1 diabetes, and mothers of children with other chronic diseases, such as asthma. Studies indicate differences in quality of life between the two.

Reviewer 2 ·

Basic reporting

The article is written in English. Literature is well referenced and relevant. The authors have used a clear text and an adequate terminology.
In my opinion, the introduction of the document should raise more accurate information regarding the different typologies that exist within diabetes. Likewise, it is important to indicate in a more detailed way sociodemographic aspects that may be directly influencing the variables of self-efficacy, social support and satisfaction in relationships.

Experimental design

Research question well defined, relevant and meaningful.
Used instruments are correct, and have high levels of reliability.
Methods are described with sufficient information to be reproducible by another investigator.
I believe that one should, on the one hand, delve more precisely the aspects related to sociodemographic variables and, on the other, make clear the reasons why the application of model 1 and model 2 offer different results (lines 210-215). It is necessary to explain what rights are taken to choose one model instead of another. If we add other measures, we must explain the reason.

Validity of the findings

Results are fascinating and easily replicable by other authors.

Additional comments

This work contains very relevant aspects that confirm the tendency or relationship between levels of self-efficacy, satisfaction relations and social support of mothers caring for children with type 1 diabetes. I suggest paying more attention the psychosocial and environmental aspects that affect the variables of Study, as well as the comparison with other groups of mothers with children with chronic diseases of particular attention.

Reviewer 3 ·

Basic reporting

See below

Experimental design

See Below

Validity of the findings

See below

Additional comments

Thank you for the opportunity to review “Quality of life of maternal caregivers of children with type 1 diabetes and the role of self-efficacy, relationship satisfaction, and social support.” This manuscript examined differences in quality of life between mothers of children with and without type 1 diabetes, as well as the role of self-efficacy, relationship satisfaction, and social support in predicting quality of life. A strength of the manuscript include the inclusion of a comparison group. A major weakness of the manuscript is the lack of inclusion of child variables, including child age. The parents’ role in management of T1D differs greatly among very young children, school age children, pre-adolescents, and adolescents. There is a developmental transition of responsibility that occurs over time, which would certainly impact parent quality of life. Inclusion of the child’s age is needed to assist with interpretation. Other child variables such as duration of T1D diagnosis would be helpful to include as well. There are a number of additional concerns, which are described below.

Introduction
1) Many of the references cited are over 10 years old - I recommend reviewing the introduction and discussion to ensure that this paper is placed in the most current context, and to highlight how this paper adds to the contemporary knowledge base.
2) Line 76. It is unclear whether the caregiver or child experiences difficulty in emotional functioning as a result of caregiver diabetes stress.
3) Line 80. Not all readers will be familiar with the transactional stress and coping model, so the authors should either briefly review the model or more clearly specify which elements provide a theoretical basis for the investigation and why.
4) Line 95. It is not clear what type of relationships the authors are referring to until later in the paragraph Given that there is a body of literature on parent-child relationships/conflict in T1D, it should be made clear that the authors are referring to marital relationships earlier on in the paragraph.
5) Line 104. It is not clear whether the authors are referring to the parent or child’s level of social support. Additionally, there is some research on social support in parents of children with T1D. The authors should review and cite this literature to expand this paragraph, which is only two sentences. See Wysocki T, Greco P. Social support and diabetes management in childhood and adolescence: influence of parents and friends. Curr Diab Rep 2006: 6: 117–122.; Lewandowski A, Drotar D. The relationship between parent-reported social support and adherence tomedical
treatment in families of adolescents with type 1 diabetes. J Pediatr Psychol 2007: 32: 427–436; Carcone AI, Ellis DA, Weisz A, Naar-King S. Social support for diabetes illness management: supporting adolescents and caregivers. J Dev Behav Pediatr 2011: 32: 581.
6) Please explain why only the quality of life mental health domain was used as the criterion variable as opposed to the total score or other subscales.
7) In hypothesis b, it seems as though the authors are only going to examine the regression in the diabetes group and it is unclear how the comparison group fits into this hypothesis.

Materials & Methods
8) In accordance with people first language (see APA Manual), the term “diabetic child” should be changed to “child with T1D” throughout the manuscript. Additionally, “diabetic group” should be modified.
9) Please specify whether there is a total score for the SF-36. Please also indicate the response format for the SF-36 and the DAS (i.e., responses range from X to Y).
10) In the description of the SF-36, it is unclear what the domain “role-physical” measures and how this differs from physical functioning. A brief explanation is needed to assist with interpretation of findings.
11) Line 151-153. It is unclear what “termed the reported health transition” means. Also, a citation noting similar research that excludes the single item is warranted.
12) The Statistical Analyses section should specific which analyses were run (e.g., t tests, regression) and why.

Results
13) The explanation of results needs further detail. It is unclear why the mother group was entered into the model when the hypothesis only specifies “in mothers of children with diabetes.”

Discussion
14) There is a growing body of literature on sleep disruption in parents of children with T1D, which could indeed be construed as having an impact on physical functioning. This issue warrants mention and interpretation.
15) The authors should offer an interpretation for why demographic factors were found to negatively impact certain variables.

---

## Round 0.2 · Minor Revisions

Reviewer 3 notes two points. I would appreciate it if you could particularly address the point that it is still unclear how the comparison group fits into the hypothesis. It is not clear if both groups are being combined to examine the impact of the stated variables on mothers as a whole regardless of whether their child has T1D

I note that two of the three reviewers recommended acceptance, and I will keep this in mind when I receive your response.

·

Basic reporting

Suitable

Experimental design

Suitable

Validity of the findings

Suitable

Additional comments

I have read the changes made, and I can say that these reflect the suggestions made by me in the evaluation process.

Reviewer 2 ·

Basic reporting

.

Experimental design

.

Validity of the findings

.

Additional comments

I have read the changes made, and I can say that these reflect the suggestions made by me in the evaluation process.

Reviewer 3 ·

Basic reporting

See below

Experimental design

See below

Validity of the findings

See below

Additional comments

The authors did a nice job addressing reviewer feedback. I have several additional suggestions for improving the manuscript.

The additional details provided in the first paragraph are helpful; however, the types of oversight and degree to which caregivers of children and adolescents with T1D provide this oversight varies with the age of the child and the duration of diagnosis. This warrants mention. Additionally several details added are already discussed further below in the introduction and warrants them unnecessary. For example, the second paragraph is now almost entirely covered in the new details added to the first paragraph. Likewise, so is the last sentence in the second paragraph on page 4.

Additional comments regarding authors' reply to initial review points.

6) Please explain why only the quality of life mental health domain was used as the criterion variable as opposed to the total score or other subscales.
REPLY: We have added an explanation at the end of the introduction.

Reviewer response to 6: This explanation is circular. The authors are hypothesizing that reduced QOL impacts caregiver ability to maintain good mental health. However, they are not measuring these things separately. The difference between quality of life – mental health and mental health in general warrants explanation as this seems to be overlooked entirely in the manuscript.

7) In hypothesis b, it seems as though the authors are only going to examine the regression in the diabetes group and it is unclear how the comparison group fits into this hypothesis.
REPLY: This mistake in the hypothesis has now been corrected; it now reads “… (b) lower levels of general self-efficacy, less relationship satisfaction, less satisfaction with social support, and fewer social support providers would be significant predictors of poorer maternal mental health in mothers based on the quality of life mental health domain.”
13) The explanation of results needs further detail. It is unclear why the mother group was entered into the model when the hypothesis only specifies “in mothers of children with diabetes.”
REPLY: Again, this mistake in the hypothesis has now been corrected; it now reads “… (b) lower levels of general self-efficacy, less relationship satisfaction, less satisfaction with social support, and fewer social support providers would be significant predictors of poorer maternal mental health in mothers based on the quality of life mental health domain.”

Reviewer response to 7 and 13: It is still unclear how the comparison group fits into the hypothesis. It is not clear if both groups are being combined to examine the impact of the stated variables on mothers as a whole regardless of whether their child has T1D. The manuscript continues to lack explanation as to why certain variables were chosen as control variables in the hierarchical regression. For example, if maternal education differed by group, then why was it not entered into the model as a control variable. Moreover, it is unclear why mother group (T1D vs. comparison) was entered as a control variable. This indicates that worse mental health was predicted by low social support, low relationship satisfaction, and low general self-efficacy regardless of mother group. However, in the discussion, the authors interpretation is unclear implies that these findings only apply to mothers of children with T1D.

---

## Round 0.3 · accepted · Accept

Well done with this last set of amendments, and thank you for your patience.